# Strategies for Potentiating NK-Mediated Neuroblastoma Surveillance in Autologous or HLA-Haploidentical Hematopoietic Stem Cell Transplants

**DOI:** 10.3390/cancers14194548

**Published:** 2022-09-20

**Authors:** Cristina Bottino, Mariella Della Chiesa, Stefania Sorrentino, Martina Morini, Chiara Vitale, Alessandra Dondero, Annalisa Tondo, Massimo Conte, Alberto Garaventa, Roberta Castriconi

**Affiliations:** 1Department of Experimental Medicine (DIMES), University of Genoa, 16132 Genoa, Italy; 2Laboratory of Clinical and Experimental Immunology, IRCCS Istituto Giannina Gaslini, 16147 Genova, Italy; 3Pediatric Oncology Unit-IRCCS Istituto Giannina Gaslini, 16147 Genoa, Italy; 4Laboratory of Molecular Biology, IRCCS Istituto Giannina Gaslini, 16147 Genova, Italy; 5Department of Pediatric Hematology/Oncology and HSCT, Meyer Children’s University Hospital, 50139 Florence, Italy

**Keywords:** natural killer cells, neuroblastoma, immunotherapy, haplo-HSCT, tumor escape, inhibitory axes, exosomes

## Abstract

**Simple Summary:**

High-risk neuroblastomas (HR-NB) are malignant tumors of childhood that are treated with a very aggressive and life-threatening approach; this includes autologous hemopoietic stem cell transplantation (HSCT) and the infusion of a mAb targeting the GD2 tumor-associated antigen. Although the current treatment provided benefits, the 5-year overall survival remains below 50% due to relapses and refractoriness to therapy. Thus, there is an urgent need to ameliorate the standard therapeutic protocol, particularly improving the immune-mediated anti-tumor responses. Our review aims at summarizing and critically discussing novel immunotherapeutic strategies in HR-NB, including NK cell-based therapies and HLA-haploidentical HSCT from patients’ family.

**Abstract:**

High-risk neuroblastomas (HR-NB) still have an unacceptable 5-year overall survival despite the aggressive therapy. This includes standardized immunotherapy combining autologous hemopoietic stem cell transplantation (HSCT) and the anti-GD2 mAb. The treatment did not significantly change for more than one decade, apart from the abandonment of IL-2, which demonstrated unacceptable toxicity. Of note, immunotherapy is a promising therapeutic option in cancer and could be optimized by several strategies. These include the HLA-haploidentical αβT/B-depleted HSCT, and the antibody targeting of novel NB-associated antigens such as B7-H3, and PD1. Other approaches could limit the immunoregulatory role of tumor-derived exosomes and potentiate the low antibody-dependent cell cytotoxicity of CD16 dim/neg NK cells, abundant in the early phase post-transplant. The latter effect could be obtained using multi-specific tools engaging activating NK receptors and tumor antigens, and possibly holding immunostimulatory cytokines in their construct. Finally, treatments also consider the infusion of novel engineered cytokines with scarce side effects, and cell effectors engineered with chimeric antigen receptors (CARs). Our review aims to discuss several promising strategies that could be successfully exploited to potentiate the NK-mediated surveillance of neuroblastoma, particularly in the HSCT setting. Many of these approaches are safe, feasible, and effective at pre-clinical and clinical levels.

## 1. Introduction

During embryo development, cells of the neural crest migrate through a process of epithelial to mesenchymal transition and give rise to sympathoadrenal precursors differentiating into peripheral nervous cells of the sympathetic ganglia and catecholamine-secreting chromaffin cells of the adrenal medulla. These are the sites where most frequently the lack of precursors differentiation results in the generation of neuroblastoma (NB), a solid tumor of the childhood [1,2]. Different genetic alterations have been detected that may induce and/or support tumor growth. These include: (i) Amplification of MYCN on 2p24, a master regulator that sustains the NB cells growth, and repress their differentiation, thus being associated with an unfavorable prognosis [3]; (ii) gain of function of ALK, also located at 2p, whose expression is limited to neural tissues and is considered a predisposition gene being involved in the majority of familial NB and 10–15% of sporadic ones [4]. ALK cooperates with MYCN, activates the MAPK pathways, often dysregulated in relapsing NB, and upregulates the proto-oncogene RET; (iii) overexpression of Lin28B, which negatively regulates miRNA biogenesis by depleting the let-7 family of miRNA, and affects the stability of the AURORA kinase; (iv) loss of 1p and 11p where still undefined tumor suppression genes are located.

NBs are extremely heterogeneous ranging from very low/low-risk tumors, often spontaneously regressing and requiring a clinical follow-up only, to high-risk tumors (HR-NB), detected in approximately 60% of cases and characterized by metastasis at different sites, particularly the bone marrow (BM) and the bone, resistance to therapy, and high incidence of tumor relapse [1,5]. In particular, HR-NBs can be defined by the presence of metastatic disease in children older than 12/18 months or by the MYCN amplification in patients of any age. HR-NB patients are treated with a very aggressive therapeutic protocol that in the maintenance terminal phase includes the anti-GD2 immunotherapy.

Current therapy showed benefits with a 5-year overall survival of up to 50%. However, it is associated with short and/or long-term side effects; these include neurological symptoms, renal, pancreatic, and thyroid dysfunction, poor growth, abnormal pubertal progression, infertility, and, for long-term survivors, increased risk of secondary solid or hematological malignancies. Moreover, approximately half of HR-NB patients do not respond to therapy and relapse [6,7]. Treatment of relapsing patients includes additional chemotherapeutic drugs, ALK inhibitors in the case of ALK-positive NB, and the use of Iodine-131 labeled MIBG, a molecule similar to noradrenaline avidly captured by NB through the noradrenaline transporter. Most recent approaches move to the personalized targeting of mutated or dysregulated molecules mimicking the strategy already adopted for other malignancies. Moreover, treatments switching off molecular pathways fueling NB growth as, for example, inhibition of telomerase or the AURORA kinase, which was reported to disrupt MYCN [8,9], are under investigation. Unfortunately, despite the use of different, sometimes compassionate or off-label, therapeutic approaches, relapsing patients have very poor survival probability highlighting the urgent need for more effective treatments.

The understanding of the in vivo effectiveness of immunotherapeutic approaches is still challenging. Immunotherapy represents one among several interventions characterizing the combined therapy of HR-NB patients, treatments that often require modifications when adverse events occur. Moreover, to date, patients enrolled for innovative immunotherapeutic approaches have been already treated with several drugs, thus making it difficult to extrapolate the effectiveness of each intervention. Due to the high complexity of the results interpretation, a multidisciplinary team should be involved to set the best-personalized approach, taking into consideration basic, preclinical, and clinical evidence. 

One of the main goals of the immunotherapeutic phase in HR-NB should be represented by the control of the BM disease. Patients mainly relapse in BM and fatally progress with BM-infiltrating NB cells characterized by very low or null expression of HLA class I molecules (HLA-I) [10]. The nature of BM disease makes NB an optimal target for a cytotoxic component of the immune system represented by natural killer (NK) cells. In fact, the constitutive presence of high numbers of NK cells in BM, theoretically ensures a fruitful interaction between NK and NB that, conversely, can be hindered by the abundant fibrous extracellular matrix present in solid tumors. Moreover, due to the molecular mechanisms regulating NK cell function, the absence of HLA-I on BM-infiltrating NB cells is expected to favor strategies aimed to potentiate NK cell cytotoxicity. During the past two decades, scientists shed light on the molecular mechanisms regulating the activity of human NK cells paving the way for strategies potentiating their function in diseases such as cancer. It has been established that the function of NK cells is finely tuned by a huge amount of activating and inhibitory receptors whose specific ligands are differentially expressed in normal or pathologic cells. The activating receptors NKp46 (CD335), NKp44 (CD336), and NKp30 (CD337) (collectively termed natural cytotoxicity receptors, NCRs), NKG2D, and DNAM-1, are under the control of the inhibitory ones, the relative ratio of the specific ligands on the target influencing its susceptibility to the NK cell-mediated killing [11]. The strongest inhibition is mediated by HLA-I molecules which are highly expressed in normal cells and, interacting with inhibitory Killer Ig-like receptors (iKIRs), avoid auto-aggression. Activating counterparts of iKIRs have also been characterized, which differ in the transmembrane and cytoplasmic portions (aKIRs). iKIRs are acquired during NK cell differentiation and, by interacting with autologous HLA-A, -B, -C alleles (KIR ligands, KIR-L) [12], allow NK cell education, which leads to the acquisition of optimal cytolytic machinery. Therefore, “licensed” NK cells spare self while spontaneously killing transformed cells, which are characterized by a reduced or absent expression of HLA-I and upregulation of ligands specific for the activating NK receptors. Unlicensed NK cells are poorly functional; their function, however, can be strengthened by cytokines or the mAbs-mediated engagement of the FcγRIIIA (CD16) leading to the antibody-dependent cell cytotoxicity (ADCC) [11]. In healthy donors most circulating NK cells are licensed cells belonging to two main subpopulations distinguished on the basis of their receptor repertoire. In particular, most NK cells are CD56dim mature cells characterized by the expression of CD16 and KIRs, molecules that are absent in the residual fraction of immature CD56bright NK cells (about 10 %) showing the expression of the inhibitory NKG2A receptor. The latter, specific for the non-classical HLA-E alleles, is gradually lost in maturating NK cells acquiring KIRs expression, according to the linear model of NK cell development [13]. Immature and mature NK cells were initially described as clearly different in terms of effector functions. In particular, the CD56bright CD16neg population was described as more efficient in releasing INF-γ in response to stimulatory cytokines whereas the CD56dim CD16pos population was depicted with a more potent constitutive cytolytic activity due to the higher content of perforin and granzyme. However, although equipped with a reduced cytolytic machinery, the cytolytic activity of immature NK cells can be efficiently recovered by the use of stimulatory cytokines, and activated CD56dim cells efficiently release IFN-γ. Thus the functional dichotomy between the two NK cell populations has been partially revised [14] and contextualized. 

However, these two main circulating NK cell subsets greatly differ in their ability to support immunotherapies based on tumor-targeted monoclonal antibodies acting through the CD16 engagement. Moreover, they differ in tissue localization, a property dictated by peculiar chemokine receptor repertoires [15]. Importantly, two other relevant NK cell types have been more recently described in humans. One is represented by tissue-resident NK cells (tr-NK) that lack CD16 expression and can be identified by the expression of peculiar markers including CD69 and CXCR6. The second one is represented by the KIRpos NKG2Aneg NKG2Cpos CD57pos terminally differentiated adaptive NK cell population that has been observed during chronic stimulation (i.e., by CMV infection), and is characterized by enhanced ADCC [16]. Along this line, leukemic patients receiving αβT/B-depleted haplo-HSCT and having cytomegalovirus infection/reactivation showed a more differentiated NK-cell phenotype [17].

Neuroblastoma is a HLA-I neg/low tumor; despite this, clinical evidences indicate that in patients it is hardly controlled by the plethora of NK cell populations described above, suggesting the presence at the tumor cell surface of key inhibitory axes hampering the immune responses [18]. These include B7-H3, a surface ligand expressed by NB cells [10], that directly stimulates tumor growth, and inhibits the cytotoxic lymphocytes interacting with a still unknown inhibitory receptor [18,19]. For the latter characteristic, B7-H3 has been included in the immune checkpoint molecular family. In this context, NBs also constitutively express or upregulate the expression, under the influence of IFN-γ, of PD-Ls, ligands of the well-known PD1 inhibitory receptor [20,21]. Interestingly, also B7-H3 expression seems to be up-regulated by IFN-γ in 3D NB model [22]. Therefore, novel therapeutic approaches in NB patients aim to restore the NK cytotoxicity by blocking the immune checkpoint axes or administering in vitro chimeric antigen receptor (CAR) engineered T or NK cells specific for tumor-associated antigens such as GD2 [23,24] or B7-H3 [25].

Another phenomenon under the magnifying glass of researchers is the NB microenvironment. Most data suggest that cancer cells can shape the tissue environment impairing the function of immune effectors; for example, tumors can affect their recruitment, induce their anergy, downregulate molecules crucial for cancer recognition, or produce soluble ligand functioning as decoy molecules for activating receptors [18,26]. Examples include the modification of the chemokine milieu [15], the production of indoleamine 2,3-dioxygenase (IDO) and prostaglandin E2 (PGE2), the release of immunomodulatory cytokines such as TGF-β1 [27], the downregulation of HLA-I, a phenomenon typical in NB, and PVR, the ligand of DNAM-1 activating receptor [28]. Tumors also release microvesicles such as exosomes that carry proteins, microRNA (miRNA), and long non-coding RNA (lncRNA) affecting anti-tumor strategies including the susceptibility of NB to drugs and immune cell recognition [29].

Importantly, new transplant strategies are also under investigation. Techniques include the selection of the more suitable hematopoietic stem cells (HSC) donor considering, for example, the presence of alloreactive NK cells, i.e., NK cells subset(s) lacking iKIRs specific for HLA-I allele(s) of the recipient. This can occur in an allogeneic setting such as HLA-haploidentical hematopoietic stem cell transplant (haplo-HSCT) [30], as well as in autologous HSCT (aHSCT) where uneducated, poorly functional NK cells of the patients are rescued by the cytokines storm unleashed during the post-transplant immune responses [31].

In this review, we describe strategies aimed to potentiate the NK cell-mediated tumor surveillance in HR-NB patients during standard and novel therapeutic protocols. These include haplo-HSCT, a setting that gained attention in recent years. We also briefly mention immunotherapeutic approaches used in other malignancies, discussing how these strategies could be adapted for the treatment of HR-NB patients depending on their NK and NB immunophenotypic landscape.

## 2. Standard Immunotherapy for HR-NB Patients

The current standard therapeutic approach for HR-NB patients consists of an intense treatment protocol including: (i) An induction chemotherapy with multiple chemotherapeutic drugs such as cisplatin, vincristine, carboplatin, etoposide, and cyclophosphamide combination (COJEC); (ii) a surgical tumor resection, if possible without serious complications; (iii) high-dose chemotherapy; (iv) radiotherapy; (v) a maintenance phase with autologous HSCT (aHSCT) followed by immunotherapy with the anti-GD2 mAb plus isotretinoin (Figure 1).

Different anti-GD2 mAbs have been utilized in patients depending on the country of the clinical study (see below). However, all the antibodies used have an Fc portion capable of activating cells of the immune system expressing the FcγR such as NK and myeloid cells. For these reasons in some countries, GM-CSF has been added to the infusion of anti-GD2 antibodies. Adding the anti-GD2 antibodies in the maintenance phase significantly improved survival rates, although a better understanding of the limitations and opportunities of the antibody-based immunotherapy is important to improve the efficacy. 

Ganglioside GD2 is a glycolipid strongly expressed at the surface of most NB cells, while its expression in normal tissues is limited to neurons, mature peripheral pain nerves, and skin melanocytes. The identification of GD2 expression on NB cells can be dated back to the 1980s [32] and a few years later, the first mAb against GD2, the murine antibody m3F8, was generated [33]. This mAb showed anti-NB activity, was tolerable, but also induced side effects such as hypertension, fever, urticarial reaction, and the generation in patients of human anti-mouse antibodies (HAMA) [34] leading to the development of chimeric or humanized mAbs. However, some patients still develop neutralizing human anti-chimeric antibody (HACA) [35] or anti-humanized antibody (HAHA) [36].

In 2010, Yu and colleagues of the Children’s Oncology Group (COG) [37] reported on a randomized Phase 3 trial of the chimeric dinutuximab (ch14.18, tradename Unituxin) in newly diagnosed children with HR-NB. All children received induction chemotherapy, surgery, local irradiation, and autologous HSCT; then, patients were randomized to receive either five cycles of dinutuximab (3 in combination with GM-CSF and 2 with IL-2) and cis-retinoic acid (CRA) or CRA alone, a product favoring NB differentiation [38,39]. The results showed that the dinutuximab arm had significantly higher overall survival (OS) (2-y OS; 86% vs. 75%) and event-free survival (EFS) (2-y EFS; 66% vs. 46%) rates compared with CRA alone. Based on this result, definitive evidence for the effectiveness of anti-GD2 immunotherapy was demonstrated to become the new standard of care for maintenance therapy in children affected by HR-NB post aHSCT. Owing to these data, NB became the first pediatric solid tumor for which immunotherapy is an established therapeutic modality given the approval of the anti-GD2 antibody. In fact, in 2015, both FDA and EMA approved dinutuximab in combination with GM-CSF, IL-2, and CRA after aHSCT for the treatment of pediatric patients with HR-NB who achieved at least partial response (PR) with prior first-line therapy.

The long-term follow-up of the same cohort confirmed superior 5-year EFS and OS for newly diagnosed patients with HR-NB randomized to immunotherapy compared with those randomized to CRA alone (EFS: 57% vs. 46%, *p* = 0.042 and OS: 73% vs. 57%, *p* = 0.045) [40]. The genotypic analysis of the NK cell population in this cohort of patients identified a subset of patients, carrying KIR2DL2/C1 and KIR3DL1/Bw4 alleles who had clinical benefit with immunotherapy as compared with isotretinoin alone. This result, however, is in contrast with four prior published studies that analyze KIR/KIR-L genotypes in NB patients receiving anti-GD2 mAb therapy and reported a better outcome in KIR/KIR-L mismatch patients [31,41,42]. The reasons for this discrepancy are relevant for the definition of the criteria for the best donors’ selection in allogenic HSC transplants and will be discussed later.

Coming back to the benefit for patients receiving immunotherapy, it should not be disregarded that the survival decreases over time due to late relapses; thus, one of the challenges is to discover strategies to increase the response rate and prevent late relapses with a reduced therapy-related toxicity. Along this line, the benefits vs. toxicity of IL-2 administration was addressed in two main clinical studies. In two non-randomized cooperative German NB Trials (NB90 and NB97), dinutuximab was administered without cytokines after aHSCT. Patients did not have improved EFS, but OS was superior; 3-y OS was 68%, 57%, and 47% for patients receiving dinutuximab, maintenance chemotherapy, or no further post-transplant therapy (due to either refuse or other reasons, control group), respectively [43]. The International Society of Pediatric Oncology European Neuroblastoma (SIOPEN) group subsequently conducted an open-label, phase 3 randomized clinical trial in which dinutuximab beta (tradename Qarziba) was given as maintenance therapy with or without the concomitant use of subcutaneous IL-2 [44]. The addition of high dose of IL-2 (doubled compared to the dose used in the COG trial; 8 h of infusion for 5 consecutive days) to anti-GD2 mAb did not increase EFS or OS and was associated with a higher termination rate and increased toxicity, including fever, pain, allergic reaction, capillary leak syndrome, gastrointestinal problems, hematological toxicity, and central neurotoxicity. Based on these data, IL-2 administration was eliminated in the European HR-NB Study 2 (HR-NBL2).

The SIOPEN group also analyzed its immunotherapy cohort (2009 to 2013, *n* = 378) with the historical control group, who received standard treatment alone (2002 to 2009, *n* = 466) [45]. It was found that the anti-GD2 cohort had significantly better EFS and OS. Multivariate analysis showed that no immunotherapy, incomplete response before immunotherapy, and the involvement of more than one metastatic site at diagnosis were significant risk factors for relapse/progression. However, no analysis of the immune compartment was performed in this cohort of patients making it impossible to correlate survival with any property of cells potentially engaged by the immunotherapy and including macrophages and NK cells. On the contrary, some features with prognostic value have been purposed analyzing immune cells in patients receiving a long-term continuous infusion of anti-GD2 mAb to improve the tolerability of dinutuximab beta. This study showed a decreased incidence and intensity of neuropathic pain [46] and demonstrated, on day 8 after the start of mAb infusion, a superior event-free survival (EFS) and a higher level of ADCC in patients having high-affinity FcγR2A and 3A genotypes, compared to those with low-affinity FcγR genotypes. This result points out that ADCC benefits likely rely on the cooperation of the NK and myeloid compartments. In fact, myeloid cells including macrophages and dendritic cells, release upon mAb-mediated engagement a plethora of cytokines (IL-12, IL-15, IL-18 and IL-23) acting in synergy and stimulating the main NK cell effector functions [47,48,49]. An advantage in terms of survival has been also observed for patients with haplotype B carrying activating KIRs (with the exception of KIR2DS4). In particular the highest benefit of the anti-GD2 immunotherapy was observed in patients showing a combination of haplotype B and high-affinity FcγRs, with a correlation among the presence of KIR2DS2, high ADCC levels, and prolongation of EFS. Despite genotypic information, however, no data have been provided about the surface expression of FcγRs or other activating receptors involved in the killing of NB, and the size of the FcγRpos population such as NK cells at the time of immunotherapy make it impossible to carry out any correlation between the effectors’ landscape and efficacy of immunotherapy. Conversely, although in a more limited cohort, Nassin and colleagues evaluated, at the time of immunotherapy and beyond, the immune reconstitution following aHSCT in HR-NB patients receiving various conditioning regimens [50]. This analysis showed that all patients, regardless of the conditioning received and the presence/absence of residual disease at the time of aHSCT, have an abnormal high percentage of CD16neg NK cells. Moreover, patients with residual disease at the time of transplantation showed low absolute counts of lymphocytes and CD16pos CD56neg cells, indicating that, at time of immunotherapy, the immune system was not fully reconstituted limiting dinutuximab efficacy.

## 3. Paving the Road for Allogeneic Hematopoietic Stem Cell Transplants in NB: A Lesson from Pediatric Hematological Malignancies

Allogeneic HSCT has become an effective therapy to cure several hematological malignancies, including high-risk leukemias, in both adult and pediatric patients [51]. Its curative potential relies on the killing of patients’ tumor cells by donors’ immune cells, mainly T and NK cells that reconstitute after transplant. The availability of HLA compatible donors, however, is limited and found for 45–75% of patients only in the Caucasian population or even less in other ethnic groups [52]. This prompted the development of haplo-HSCT, i.e., a transplant from a donor having one identical HLA haplotype and the other mismatched. For pediatric patients haplo-HSCT has revealed an attractive and feasible option since parents represent easily and quickly available HLA-haploidentical donors [53]. Accordingly, to potentiate the NK-mediated anti-tumor control in NB patients with a worse prognosis, the autologous setting starts to be substituted over time by strategies addressing the feasibility and safety of haplo-HSCT, with or without the infusion of donor-derived NK cells (Table 1).

To avoid fatal complications in such HLA-incompatible settings, a first successful strategy was designed, based on the infusion of “mega-doses” of highly purified CD34pos HSC infused after an intensified myeloablative conditioning, to prevent graft rejection, and an extensive T-cell depletion to avoid graft-versus-host disease (GvHD) [54]. In this T-cell depleted haplo-HSCT, NK cells were the first lymphocyte population emerging after transplant and revealed their crucial contribution against tumor recurrence. In particular, the seminal studies from the Ruggeri group [55] demonstrated the relevance of the presence in the donor of alloreactive NK cells, i.e., NK cells with a iKIR/KIR-L mismatch in graft vs. host direction. In particular, in adult acute myeloid leukemia (AML) patients who received a graft from iKIR-mismatched haploidentical donors, the NK cell recovery protected from leukemia relapse with a 5-year survival probability of around 60% (vs. 5% in non-alloreactive haplo-HSCT). NK alloreactivity played a relevant role also in pediatric patients receiving the same CD34pos haplo-HSCT, with an overall survival rate very high in case of high-risk acute lymphoblastic leukemia (ALL) (~70% survival vs. 35% in non-alloreactive transplants), although less impressive in AML patients (~40% vs. 20% in non-alloreactive transplants) [56,57]. Importantly, besides preventing leukemic relapse, donors’ alloreactive NK cells also contribute to killing both T lymphocytes and antigen-presenting cells of the recipients, reducing graft rejection and GvHD [58]. In addition, a critical role in preventing leukemia relapse and viral infections was suggested for NK cells expressing activating KIRs (aKIRs) [59] that, differently from iKIRs, transduce positive signals via adaptors molecules. To date, except for KIR2DS1 recognizing HLA-C2, limited data are available on ligand recognition by the other aKIRs [60,61]. Interestingly, reconstituting NK cells expressing KIR2DS1 and lacking KIR2DL1, HLA-C2 specific KIRs with opposite function, efficiently eliminated HLA-C2pos leukemic cells, thus likely contributing to the higher survival rate upon CD34pos haplo-HSCT [62].

These results indicated that an accurate donor selection was of paramount importance to fully exploit the anti-tumor potential of the NK cell populations emerging after transplant [57,63]. In this context, an expansion of a peculiar NK cell subset in HSCT patients, driven by cytomegalovirus (CMV) infection/reactivation, has been detected that correlated with reduced leukemia relapse rates [64,65,66,67]. The CMV-induced NK cell subset is hallmarked by the expression of the HLA-E-specific activating receptor NKG2C, the highly differentiated surface signature CD57pos KIRpos NKG2Aneg, unusual longevity, broad epigenetic modifications, and altered signaling molecules expression that enhance their ADCC potential [16]. CMV-induced NK cells are referred to as adaptive or memory NK cells, frequently lack the signaling molecule FcεRγ, kill more efficiently IgG-coated targets, and also exhibit robust expansion via CD16 engagement [68]. It is conceivable that the presence of adaptive NK cells could increase the efficacy of therapeutic mAbs favoring the NK-mediated killing of tumor targets [16,69]. In view of these specialized features, adaptive NK cells hold translational promise in novel immune therapies in different pathological settings including NB.

It is of note, however, that in CD34pos haplo-HSCT patients the generation of fully mature and cytolytic NK cell populations, including alloreactive and adaptive NK cells, requires several weeks from HSC infusion [57]. Indeed, the first wave of NK cells after HSCT is represented by immature CD56bright NKG2Abright CD16neg/low NK cells, while the more differentiated CD56dim CD16pos KIRpos NKG2Aneg NK cells, possibly containing alloreactive and adaptive NK cells, emerge later; this disequilibrium favors tumor relapses and virus infections/reactivations. To reduce the gap in which patients are immunodeficient, a novel approach has been developed based on the HSC mobilization in peripheral blood and selective removal of αβT and CD19pos B cells, to avoid GvHD and EBV-related lymphoproliferative disorder [70]. In αβT/B-depleted haplo-HSCT the graft contains high numbers of donor-derived CD34pos HSC, mature NK cells, and γδ T cells (median 34.6 × 10^6^/kg and 8.1 × 10^6^/kg respectively) that, promptly available, could prevent early leukemia relapses and virus reactivation/infection [56,71]. αβT/B-depleted haplo-HSCT had successful clinical results with a high rate of leukemia-free survival, low rate of graft failure, and low risk of severe GvHD, without the need for post-transplant GvHD prophylaxis, which may impair innate immunity-mediated graft vs. leukemia (GvL) effects [17,72]. At variance with purified CD34pos HSCT, in donors’ αβT/B-depleted haplo-HSCT the NK alloreactivity had a negligible impact on children’ leukemia-free survival rate [17,56]. This observation could be explained by the high success rate of this transplant procedure (more than 70% leukemia-free survival on average), by the presence of additional anti-tumor effectors (including γδT and NK-T), and by the optimization of donor selection criteria that make difficult to determine direct correlations with specific biological traits. Indeed, the donor selection process required complex molecular and phenotypical analyses to identify the presence of NK alloreactivity, the KIR B genotypes to determine the presence of at least a set of KIR genes including aKIRs, the preferential choice of the mother, since mother-derived grafts showed better alloreactive effects [73], and the evaluation of the surface expression levels of activating receptors on donor NK cells, namely NKp46 that is involved in leukemia killing, and NKG2C, thus avoiding NKG2Cneg/neg donors [17,56,74]. This careful selection together with the presence of other effectors probably obscure the impact of NK alloreactivity. A negative effect exerted on mature NK cells by myeloid-derived suppressor cells contained in the αβT/B-depleted graft must also be considered [75].

Another haplo-HSCT setting consists of an unmanipulated haplo-HSCT followed by high dose post-transplant cyclophosphamide (PTCY) at early time points (on days +3 and +4). The drug acts on proliferating cells and mediates in vivo depletion of both recipient’s and donor’s alloreactive T cells [76]. This strategy is an interesting option, considering that it is easier and less expensive compared to αβT/B-depleted haplo-HSCT and could be easily applied in developing countries. The effectiveness of haplo-HSCT with PTCY, however, has been largely investigated in adults affected by leukemia and other hematological diseases [72], but only partially studied in children [77,78]. Moreover, PTCY administration was shown to induce NK cell depletion, also leading to the elimination of alloreactive NK cells possibly contained in the graft and undergoing proliferation in response to high IL-15 levels measured in patients’ sera [66]. Thus, further studies are required to better understand the impact of this transplant setting on anti-tumor NK cells properties.

## 4. Haploidentical Hematopoietic Cell Transplantation in HR-NB Patients: Where We Are

Due to the success of haplo-HSCT in hematological malignancies, this strategy started to be explored in various solid tumors including NB (Table 1). Pilots studies with HSC-enriched transplants observed the association between decreased tumor relapses and the presence of donor vs. recipient iKIR/KIR-L mismatch, suggesting a possible association between NK cell alloreactivity and tumor control [79,80,81]. Thus, the therapy administrated to HR-NB patients has been integrated over time with grafts from haploidentical donors that, in the case of Tαβ/B cell-depleted grafts are highly enriched in donors’ NK, γδT, and myeloid cells; in addition, the patient could be infused with in vitro activated mature NK cells from the donors. In recurrent/refractory HR-NB patients, the infusion of “haplo products” has been followed by humanized [82] or murine [83] anti-GD2 antibodies. These studies, involving 13 and 35 patients, respectively, suggested the feasibility and efficacy of haplo-approaches in the treatment of HR-NB patients with minimal residual disease. This result was confirmed by Lang and colleagues who greatly contributed to the development of haploidentical transplantation in patients with solid tumors. In particular, αβT/B cell-depleted grafts were used in children affected by 17 solid tumors in remissions including NB [84]. The study explored the reconstitution of the NK cell compartment providing evidence that on day 14 post-transplant the absolute NK cell number was significantly higher than that observed in a cohort of 42 patients who received CD34pos enriched grafts from mismatched donors. Conversely, no significant differences were found at time points covering the 21–90 days post-transplant window. The NK cells observed at day 14 post-transplant could be enriched with bona fide mature NK cells present in the infused graft, which contained a median NK cell number of 10^6^/kg body weight. However, the CD56dim NK cells recovered from patients at early time points showed phenotypic differences compared to classical CD56dim NK cells from healthy donors. In particular, they show high expression of NKG2A and CD62L and a reduced expression of NKG2D, characteristics compatible with cells maturating from CD56bright NK cells rather than with cells expanded from the infused donors’ mature CD56dim NK cells. Importantly, in transplanted patients, 63% of the CD56dim NK cells express CD16. Accordingly, NK-mediated ADCC against NB cells was observed in the presence of the humanized anti-GD2 mAb, both in the absence and in the presence of IL-2. Interestingly, since day 14 post-transplant, a certain degree of Ab-independent natural cytotoxicity against NB cells was also appreciated.

A joint effort by the German and Sweden groups provided additional information with two prospective phase I/II studies analyzing 26 HR-NB patients with refractory or relapsed disease receiving haplo-HSCT [30]. These studies reported engraftment in 96% of patients with an absence of transplant-related mortality (TRM), although all patients were previously subjected to a combined and intensive therapy according to national and international protocols. No increased viral infections were observed, although T-cell recovery was delayed compared to autologous transplants or from matched sibling donors. This effect could be ascribed to the rapid reconstitution of the NK cell compartment together with good neutrophil recovery, as well as the rigorous pharmacologic prophylaxis and monitoring of viral reactivation. Fifteen of the patients either had no GVHD or developed grade I, while 11 patients had grade II or III. The analysis included 15 patients who developed GVHD after donor’s T lymphocyte (DLI) infusion. The GVHD, however, was not associated with a superior benefit in terms of 5-year EFS compared with relapsing patients who underwent a second course of high-dose chemotherapy and aHSCT (19% versus 15%). The EFS was higher than that (6%) reported by the Center for International Blood and Marrow Transplant Research (CIBMTR) in allogeneic matched HSCT [85]. The benefits clearly correlated with the presence of a reduced tumor burden, more evident in patients undergoing haplo-HSCT in complete remission. No differences have been observed in patients having a different absolute number of NK cells at day 30 after transplantation or between patients having or not iKIR/KIR-L mismatched NK cells. These data suggest that the natural NK cell cytotoxicity arising at early time points in the haplo-HSCT setting is not potent enough to control tumor progression in the absence of a pre-transplant complete remission. The reason may rely on the presence of suppressor cells in the infused graft or a donor selection still to be optimized particularly in relation with the NB immunophenotypic properties, aspects discussed below.

Having demonstrated the safety of haplo-HSCT, this technique has been combined with the infusion of chimeric anti-GD2 mAbs and IL-2 in children with relapsed HR-NB [86]. The immune monitoring of this cohort of patients provided some very relevant information. First, the early presence of NK cells with a peak of CD56pos CD16pos cells at 14 days post-transplant has been confirmed. Importantly these NK cells show high ADCC capability [79,84]. The schedule and concentration of the anti-GD2 antibody were adapted to maintain clinically relevant serum levels, the onset of human anti-chimeric antibody (HACA) being detected in one patient only. The chimeric antibody demonstrated the ability to engage, activate, and expand patients’ CD56pos CD16pos NK cells that increased the expression of CD69, a typical NK cell activation marker. Moreover, peripheral blood NK cells of patients at 60 days post-transplant have been shown, ex vivo, to kill different NB cell lines upon engagement by dinutuximab. The levels of Fc- and complement-dependent cytotoxicity clearly correlated with the antibody concentration as well as with the surface density of GD2 on NB cells. This result could be predictable from an immunologic point of view, due to the isotype of the antibody (IgG1) and the described expression of GD2 on NB; however, we must take into account the existence of GD2neg/low NB variants, which were recently identified by our and other groups using different methodological approaches [87,88]. While GD2 expression on target cells had a significant impact on the ability of dinutuximab to activate NK cells, differences in terms of KIR/KIR-L genotype or CD16 allotypes did not appear to correlate with differences in patients’ EFS or OS.

## 5. Exploring the NK/NB Molecular Interactions to Optimize Haploidentical Transplant

The data discussed above indicate that the haplo-approach needs to be optimized to obtain a superior benefit from cytotoxic NK cells contained in the infused cell products that are engaged during immunotherapies. This could be achieved adopting different strategies including an accurate, NB-specific selection of the optimal NK cell haplo-donor, and the strengthening of the anti-tumor NK cell functions (Figure 2). 

Therapeutic strategy cannot disregard the deep difference existing in terms of biological and immunophenotypic features between hematologic malignancies and NB. Differently from hematologic malignancies, NB are HLA-I neg/low tumors due to defects in proteasomal and TAP proteins [10,89], hypothetically representing optimal targets for NK-mediated killing. However, tumor cells purified from BM aspirates of HR-NB patients showed up to ten-fold lower susceptibility to NK-cell-mediated killing compared to NB cell lines [28]. The reduced killing was partially restored by mAb-mediated masking of B7-H3, a tumor antigen heterogeneously expressed by primary tumors but highly expressed by all the BM-infiltrating NB analyzed [10,87,90]. After very promising preclinical results, B7-H3 has been targeted in clinical trials in patients suffering different kinds of tumors (https://www.clinicaltrials.gov (accessed on 3 August 2022)). Functional data show that a receptor(s) with inhibitory function can be expressed by NK and T cells; however, it has not been identified yet, thus making not practicable a selection of haplo-donors based on its expression [91,92]. Beside the constitutive expression of inhibitory ligands such as B7-H3, IFN-γ-inducible inhibitory axes exist as mechanisms of tumor adaptation to the immune pressure. HLA-I and PD-Ls molecules are heterogeneously expressed by MYCNampl and non-amplified NB cell lines and show a rapid kinetic of induction/upregulation by IFN-γ [20]. Importantly, as already pointed out for B7-H3 molecule, HLA-I and PD-L1 show a variable expression in primary tumors [93]; on the other hand, they were both upregulated by IFN-γ in vitro and in vivo in a mouse model [20]. It is of note that the anti-GD2 immunotherapy can induce PD-L1 expression in NB cells and, in preclinical model, the combined treatment with anti-GD2 and anti-PD1 showed a synergic therapeutic effect [94]. The same combining therapy administrated to two patients with refractory NB led to either a complete remission or a very good partial response [21]. Prospective trials in a larger number of patients are needed to further confirm the role of this therapeutic approach.

New prospective trials should also consider the combining blockade of other constitutive or IFN-γ-inducible inhibitory axes that have been shown to limit the NK-cell-mediated immune surveillance, particularly in the first phase post-transplant where the spontaneous NK cell cytotoxicity should provide immune surveillance before anti-GD2 treatment. In this context, data on the correlation between the mRNA expression levels of the ligands of activating NK receptors and EFS and OS are not easy to interpret. A correlation between high PVR (CD155) expression and worse EFS and OS was observed in about 500 primary NB tumor [87]. Data suggest a role in vivo of TIGIT (T-cell immunoglobulin and ITIM domain) and CD96, PVR-specific inhibitory receptors counteracting the activity of the DNAM-1 activating receptor [95,96].

Although information about TIGIT expression by NK cells from NB patients are still missing, published studies in different type of cancers demonstrate that TIGIT expression is high in lymphocytes of tumor patients, thus representing a targetable inhibitory immune checkpoint receptor limiting both NK- [97,98] and T-cell mediated anti-tumor activity [18,99,100,101,102,103,104]. Importantly, the results demonstrated that TIGIT blockade was able to directly subvert the exhaustion of tumor-infiltrating NK and T cell [105,106]. Along this line, several therapeutic tools have been developed and are currently under investigation in clinical trials [103].

Coming back to the prognostic significance of other ligands recognized by NK cell receptors, also the high expression of B7-H6, ligand of NKp30 [107], had an unexpected and interesting negative prognostic value, while, the high expression of MLL5 (ligand of NKp44) [108], ULPB-2 and-3 (ligand of NKG2D) [109] correlated with better EFS and OS [87].

In NB patients, most of the data regarding the phenotypic and functional properties of NK cells have been obtained analyzing cells from peripheral blood (PB-NK). Notably, however, relapses mostly occur in the BM, where NK cells significantly differ from PB-NK. BM holds peculiar CD56bright CD16neg or CD56dim CD16low populations [110,111]. The CD56dim CD16low population has been characterized in BM of leukemia patients undergoing αβT-depleted haplo-HSCT [110] and shows similitude with a CD56dim population previously identified in BM of HR-NB patients [112]. These cells lack CX3CR1 and CXCR1 expression, resembling a TGF-β-driven phenotype [112] rarely observed in classical CD56dim PB-NK cells. A longitudinal analysis of the NK cell compartment in HSCT transplanted leukemic patients showed that the CD56dim CD16low population arises in PB and BM one month after transplant and is characterized by a great ability to produce IFN-γ in response to IL-12 plus IL-15, according to the high expression of IL-2Rβ (CD122) [110]. This population, however, shows a reduced spontaneous killing capability against tumor cells, its cytotoxicity being efficiently recovered only later, approximately one-year post-transplant. Conversely, reverse ADCC experiments showed that these cells efficiently degranulate in response to NKp46 engagement; this indicates that the degranulation defect cannot be due to deficiencies in the activating receptor signaling or degranulation machinery. If the presence of the CD56dim CD16low cells in NB patients at early phase post-transplant will be confirmed, this population could be activated, for example, by multifunctional engagers targeting NKp46 on NK cells [113] and a tumor antigen on NB cells, contributing to the control of BM residual disease. These engagers could also efficiently activate other NKp46pos NK cell population present in PB and BM of patients at early phase post-transplant; these can be represented by mature donor-derived NK cells and CD16neg immature NK cells developing from CD34pos HSC precursors, which are barely activated by the standard anti-GD2 therapy.

An additional strategy to provide a better tumor immune surveillance could rely on improving the recruitment in BM and peripheral tissues of mature CD16pos NK cells. Along this line, in NB the degree of NK and DC infiltration has a significant prognostic value [114], also supported by several studies showing a key role of DC in the activation of the NK antitumor activity [115,116,117]. Beneficial therapeutic strategies could also promote the NK cell differentiation program and/or their proliferation at the tumor site [118,119]; this may impact on the mature/immature NK cell ratio, which is highly shifted toward the immature compartment in most tumors [120]. Beside negatively regulating the cytolytic activity of NK cells [27], TGF-β deeply modifies their chemokine receptor repertoire. In particular, alone or in collaboration with IL-18, it downregulates the expression of CX3CR1 and CXCR1 while upregulating that of CXCR3 and CXCR4, a well-known BM-homing receptor [121]. Clinical grade inhibitors of TGF-β signaling have shown to potentiate the NK cell activity against NB in preclinical mouse models [122]. Further studies are needed, however, to verify the concomitant increase of NK cell infiltration at tumor sites and its relationship with the documented therapeutic effect. Importantly, these studies could shed light on the in vivo role of TGF-β in regulating human NK cell migration in peripheral tissues or their retention in BM [15].

Drugs could be used to increase the recruitment of mature NK cells at tumor sites (e.g., in the BM), including the tyrosine kinase inhibitors (TKIs) such as imatinib mesylate and nilotinib. Imatinib had beneficial effects in relapsed/refractory HR-NB patients; they received concentrations of drugs unable to directly kill cancer cells, suggesting a possible protective off-target effect of the TKI [123,124]. The analysis of the TKI effects in different leukocyte populations showed that these compounds modulate the chemokine receptor repertoire of different cells including NK cells, particularly, downregulating CXCR3 and upregulating CXCR4 [125]. This in vitro observation was confirmed by the higher levels of CXCR4 detected in NK cells and monocytes of chronic myeloid leukemia (CML) patients treated with TKI [125]. Importantly, unlike TGF-β conditioning, the TKI treatment preserved the expression of CX3CR1 and CXCR1, which are relevant for NK cell migration. Moreover, the levels of CXCR4 on NB cells were not upregulated by TKI and, after two cycles of treatments, the NKp30 mRNA levels significantly increase in BM [125]. The meaning of this last observation needs to be further explored.

## 6. Unveiling the Role of NB and NK-Derived Exosomes: How Can They Improve Immune Cell-Based Treatment Strategies

Activating and inhibitory signals regulating the NK cell mediated anti-tumor activity can be delivered by extracellular vesicles (EVs), which in NB patients have been shown loading several regulatory molecules. EVs include exosomes which are small extracellular vesicles playing a key role in cell communication that, when released by cancer cells, carry biomolecules positively regulating pro-tumoral pathways such as proliferation, angiogenesis, invasion, and immune escape [126,127]. The involvement of exosomes in tumor progression has been characterized in many cancers, including NB [128,129,130]. NB-derived exosomes carry oncogenic and metastatic signals [131,132,133], as well as microRNA molecules (miRNA) involved in chemoresistance [134,135]. It has been shown that tumor exosomes can affect the immune response by interfering with the NK cell function. The most common mechanism to accomplish this task is the exosomal expression of immunosuppressive molecules, such as TGF-β1 and ligands down-modulating the expression of receptors such as NKG2D required for NK cell activation [136]. Similarly, it has been shown that NB-derived exosomes contain the immune checkpoint ligand B7-H3 exerting immunosuppressive function against NK cells activity [131].

Nevertheless, exosomes can shape an immunosuppressive environment in multiple other ways: (i) down-regulating NK cell migration and proliferation, (ii) reducing the cytokine production (iii) transferring miRNA molecules blocking the translation of key proteins for NK cell function, and (iv) inducing metabolic alterations [137]. Notably, it was reported that tumor exosomes can express tumor antigens to sequester tumor-reactive antibodies, with the final result of hindering the ADCC mediated by NK cells [138]. Consequently, cancer-derived exosomes may strictly limit the efficacy of NK cell-based immunotherapy [139]. It has recently been demonstrated that small extracellular vesicles (sEVs) released by NB cells promote resistance to dinutuximab [140]. Liu et al. showed in a NB mouse model that sEVs released from NB cells induce dinutuximab resistance by reducing NK cell infiltration and recruiting tumor-associated macrophages (TAM) [140]. Authors demonstrated that sEVs are up taken by the spleen, where they counteract the maturation of NK cells promoted by dinutuximab, increasing the percentage of the immature NK cells subset. Furthermore, in vitro experiments showed that pre-incubation of NK cells with NB-derived sEVs inhibited the NK-elicited ADCC against NB cells [140]. Interestingly, Liu et al. propose to combine the anti-GD2 immunotherapy with tipifarnib, a farnesyltransferase inhibitor able to prevent sEVs release. The results demonstrated that such a combination would prevent the shaping of an immunosuppressive environment exerted by sEVs, ensuring a higher efficacy of dinutuximab treatment [140].

The loss of NK cytolytic activity seems to be the result of the prolonged and persistent release of immunosuppressive signals by tumor-derived exosomes. Conversely, it has been demonstrated that a short-time exposure of NK cells to cancer exosomes causes an initial up-regulation of activating receptors (NKp30 and NKG2D) and a concomitant decrease of the inhibitory ones (NKG2A), leading to a significant stimulation of NK cell function [141]. Shoae-Hassani et al. hypothesized that the short-term exposure to tumor-derived exosomes could represent an effective method to activate NK cells, amplifying their cytolytic activity against NB cells [142]. In this study, the authors showed that naive NK cells, treated with exosomes derived from NK cells previously co-cultured with NB cells, had higher cytotoxic activity against NB. In particular, NB-pre-exposed NK exosomes were able to significantly upregulate activating receptors (NKp30, NKp44, NKp46, and NKG2D) on the NK cell surface, and increase the NK-mediated production of INF-γ and TNF-α. These features conferred to NK cells higher cytotoxic effects resulting in increased NB cell death in vitro, and reduced tumor growth in vivo [142]. As reported by the authors, if the treatment with exosomes derived from NK cells previously exposed to NB cells alone is sufficient to boost NK cell cytolytic activity, the combination with a cytokine cocktail will result in a synergistic effect [142]. NB-pre-exposed NK exosomes may be instrumental to improve anti-tumor activity of NK cells and, thus, can be used as an adjuvant in immunotherapeutic treatment. Further evidence suggested the use of NK-derived exosomes coupled with immune cell therapy in NB. It has been shown that exosomes released by NK cells contain the onco-suppressive miR-186, which exerted cytotoxic effects on MYCN-amplified NB cell lines [143]. In particular, miR-186 can reduce NB cell migration and proliferation by targeting and down-regulating TGFβR1, TGFβR2, and SMAD3, thus interfering with the TGF-β-mediated inhibition of NK function. Authors delivered miR-186 mimic specifically to NK cells using nanoparticles coated with anti-CD56 antibody, showing that its ectopic up-regulation can prevent the TGF-β-mediated inhibitory effects [143]. These results suggest that the targeted delivery of miR-186 may represent a useful tool to stimulate ADCC, avoiding the TGF-β-mediated immune escape mechanism adopted by NB cells. Notably, the exosomes derived from NK cells treated with TGF-β maintained unaltered levels of miR-186. Therefore, NK exosomes retain properties even in immunosuppressive conditions further suggesting that they could be used in combination with standard immune cell-based therapies to improve NB patient outcomes [143].

## 7. Additional Immunotherapeutic Strategies and Conclusions

Several preclinical and clinical studies have been performed in the attempt to understand how the immunotherapeutic phase in HR-NB could be optimized, hopefully, leading to increased patient survival. Considering that the amount of residual disease at the time of immunotherapy could impact patients’ outcomes [144,145], recent strategies explore the benefits in children with a new diagnosis of HR-NB of anticipating the anti-GD2 treatment to the induction chemotherapeutic phase. It is the case of a single-arm phase II clinical trial, where it has been shown that the use of the humanized anti-GD2 mAb (hu14.18K322A) during the intensive induction chemotherapy is feasible, well-tolerated, and results in an early response rate of nearly 80% with no patients in disease progression during this phase [146]. Currently, COG and SIOPEN are both designing trials to confirm the feasibility of this approach. A strategy already feasible in NB patients, even in those patients who previously experienced autologous transplants, is haplo-HSCT. This transplant approach showed benefits due to a better immune landscape, especially in terms of the quality of the NK cell compartment present in the early post-transplant phase. Importantly, looking at results obtained with the haplo-HSCT approach in leukemic patients, benefits could be higher when the αβT/B-depleted transplant is used, which is characterized by high numbers of mature effectors in the final infused product including FcγRpos cells such as mature NK cells that due to their potent effector functions might better control the residual BM disease. This activity could be further potentiated by identifying optimal haplo-HSCT donors characterized by the more suitable NK cell compartment, according to the data obtained analyzing the immunophenotypic properties of NB cells in vitro, animal models, and clinical studies. The possible criteria that could be adopted for donors’ selection are summarized in Figure 2. These include as pivotal criterium the presence of a wide CD56dim CD16pos CD57pos NKG2Cpos adaptive NK cell population having a high ratio of DNAM-1pos/TIGITpos and CD96pos cells, and high levels of NCR and NKG2D. It should not underestimate, however, that the susceptibility of patients’ derived tumor cells to the NK cell-mediated killing is significantly lower than that observed for NB cell lines. Thus, the in vivo natural cytotoxicity of NK cells could be insufficient to control tumor growth. It is conceivable to consider the possibility to potentiate their function, for example, by using engineered multifunctional tools able to engage, in NK cells, CD16 and activating receptors such as NKp46 that is highly expressed at every time post-transplant (Figure 3).

These engagers could better activate, than the anti-GD2 mAb alone, CD16dim CD16neg NK cells that are abundant post-transplant, particularly in the BM. Importantly, in order to potentiate strategies based on CD16 engagement, constructs could carry inhibitor of A disintegrin and metalloprotease-17 (ADAM17), to prevent CD16 shedding and improve the efficacy of therapies [147,148,149]. Moreover, the NK vs. NB activity could be also sustained by the addition, in the engager construct, of cytokines capable to induce activation/proliferation of NK cells with low toxic side effects. Along this line, next-generation modified cytokines, mainly IL-2 and IL-15, or constructs containing these cytokines showed promising activity and safety as adjuvants in immunotherapies [150,151,152,153]. Importantly, in an autologous or haplo-HSCT, IL-15-based interventions could also favor the reconstitution of the NK cell compartment [154,155,156,157,158] increasing the effectiveness of ADCC in early phases post-transplant. It should not be underestimated, however, a recent report by Miller JS and colleague describing an in vivo detrimental effect of the systemic administration of engineered IL-15 (IL-15/N-803) in relapsed/refractory acute myeloid leukemia (AML) patients receiving haplo NK cells [159]. In fact, in two independent clinical trials (NCT03050216 and NCT0189), it has been shown that in comparison with IL-2, systemic IL-15 has a late negative effect on the in vivo persistence of adoptively transferred haplo NK cells, although causing an increased number of NK cells at early timepoints. The progressive reduction of NK cell population seems to be ascribable to rejection by CD8+ T cells of the recipient, whose frequency significantly increased after IL-15 infusion as compared to IL-2 treatment. According to this hypothesis, in vitro mixed lymphocyte reaction (MLR) experiments showed that IL-15/N-803 induces a strong responder CD8 T-cell activation and proliferation, leading to a high killer activity against stimulator-derived memory-like NK cells. To further underline the complexity of possible IL-15-based interventions, high doses of IL-15 induce, in NK cells, the activation of CIS (cytokine-inducible SH2-containing protein, encoded by *Cish*), an intracellular inhibitory checkpoint limiting their activity over time. Preclinical studies showed that strategies aimed at blocking CIS to obtain a more potent antitumor NK-mediated surveillance were effective and safe and were not paralleled by a dangerous, dysregulated NK cell expansion [160,161]. Moreover CIS blockade also improves the efficacy of additional therapeutic tools such as CAR-engineering PB-NK cells [162] and inhibitors of the TGF-β pathway [163].

The challenging achievement of an optimal NK cell activation could render NK cells less susceptible to the various constitutive and/or inducible inhibitory signals generated by NB cells [18,164]. For example, the NK cell killing of NB could be potentiated in the various phases post-transplant, particularly the initial phase, by the use of products targeting TGF-β1 that causes the dysfunction of NK cells [27] or blocking inhibitory axes including B7-H3R/B7-H3, and TIGIT/or CD96/PVR [95]. Moreover, the presence of NKG2Ahigh NK cells post-haplo-HSCT transplant together with the possibility of NB cells to upregulate HLA-I and PD-Ls in response to IFN-γ, unleashed by different immune cell types during tumor immune response, give a rationale to the combined blocking of these inhibitory axes [165]. In support of these strategies, in different types of cancers the NKG2A blockade, combined with other immunotherapeutic approaches, showed encouraging clinical results in terms of both efficacy and limited adverse events [166,167]. About the possible efficacy of PD-1/PD-Ls blockade, however, it cannot be disregarded the reduced constitutive PD-1 surface expression observed in PB- and BM-NK cells of NB patients [20]. Thus, further studies on the regulation of PD-1 expression, particularly during the course of therapy, could clarify the contribution of NK cells in the positive results obtained by blocking the PD-1/PD-Ls axes in NB patients [21].

Interestingly, all the above-depicted strategies could also interfere with the inhibitory signals carried by tumor exosomes. The strategies potentiating NB control by NK cells in haplo-HSCT could be more effective if accompanied, post-transplant, by the infusion of mature NK cells purified from the same donor, a safe strategy already experienced in the treatment of NB patients [82,168,169], Table 1 Importantly, NK cells can be infused as naïve cells, followed by anti-GD2 immunotherapy, or shortly stimulated in vitro before their infusion using different strategies including the use of IL-12 and IL-18. This cytokine combination has been shown to induce in NK cells a cytokine-induced, memory-like phenotype (CIML) responsible, in vivo, for a rapid 10- to 50-fold NK cell expansion. Moreover, CIML-NK cells show a transcriptional, epigenetic, and metabolic reprogramming responsible for their high effector functions [170,171,172]. CIML-NK or NK cell activated by using different strategies [151,173] could be also engineered with chimeric antigen receptors (CARs) specific for GD2 [174] or alternative tumor-associated molecules including B7-H3 whose targeting have already shown excellent preclinical data [25,175,176]. Importantly, different clinical trials in adult and pediatric tumor patients are ongoing to prove the safety and efficacy of targeting B7-H3 using CAR-engineered cells or humanized antibodies conjugated or not with tumoricidal compounds (https://www.clinicaltrials.gov (accessed on 3 August 2022)). Other promising CAR targets exist in NB patients including glypican-2 (GPC2) [177,178] and CD171 [179]. Besides T cells, NK cells represent only one of the alternative cell platforms for CAR; indeed, NKT cells, γδ T cells, DC, macrophages, regulatory T cells (Treg), and B cells are also analyzed [180]. NK-CARs [181] may be more convenient compared to the most common T-CAR approach. Benefits include the possibility to use haplo healthy donors as a source of NK cells without significant GVH, and the constitutive surface expression of activating NK receptors (NCR, NKG2D, DNAM-1) recognizing tumor antigens and co-operating with CARs [19,182]. All the available CAR-based platforms, some of which are designed to favor CAR-cell persistence and tumor invasiveness [183,184,185,186], have the risk of exacerbating the GVHD [187]. This event, however, could be controlled with approaches including the use of CAR construct holding suicide genes [188] or the infusion of ex vivo expanded donor-derived regulatory lymphocyte subpopulations [72]. 

To conclude, to date we have a plethora of promising interventions that can be applied to optimize immunotherapy in HR-NB patients in an attempt to prolong their survival and improve their quality of life. It is mandatory to conduct clinical trials enrolling high numbers of patients to reach statistically significant results and organize multidisciplinary and multicentric round tables to discuss the biological and clinical information available for each patient and select the most promising treatment option.

## Figures and Tables

**Figure 1 cancers-14-04548-f001:**
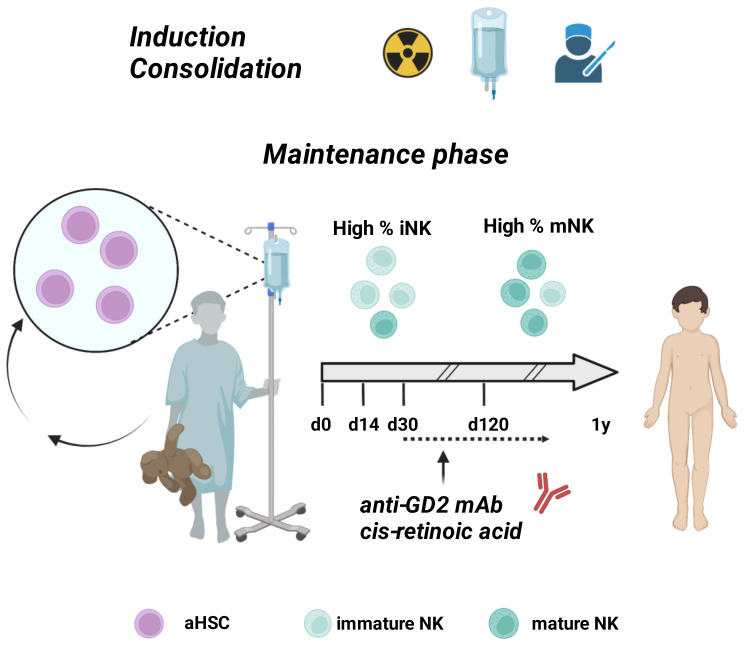
Standard therapy for patients with HR-NB. After chemo-, radio-therapy, and surgical intervention, the maintenance phase consists of autologous HSCT followed by the anti-GD2 immunotherapy starting from about 30 days post-transplant. At the time of immunotherapy NK cells still have a CD16neg/low KIRneg NKG2Abright immature phenotype (iNK), acquiring only later the CD16pos KIRpos NKG2Aneg mature phenotype (mNK). aHSC, autologous hemopoietic stem cells. Created with BioRender.com.

**Figure 2 cancers-14-04548-f002:**
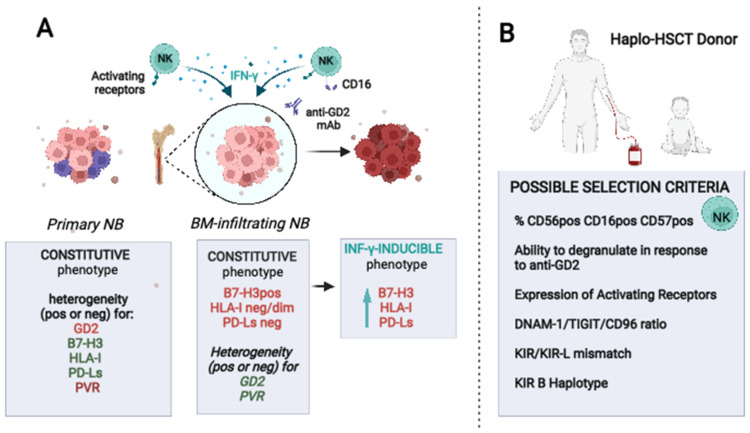
Constitutive and IFN-γ-induced immunophenotype of primary and BM-infiltrating NB cells. (**A**) Primary NB show intratumor molecular heterogeneity whereas BM-infiltrating NB cells have a more homogeneous B7-H3pos and the neg/low expression of HLA-I and PD-Ls. Inhibitory ligands are induced/upregulated by IFN-γ released upon NK cell activation by tumor ligands or the anti-GD2 mAb. (**B**) Possible features of the NK cell population to be considered for the selection of the best haplo-HSCT donor (in order of priority). Optimal selection criteria could improve the NK cell persistence and NK vs. NB cytotoxicity, both spontaneous and anti-GD2 mAb-mediated. Created with BioRender.com.

**Figure 3 cancers-14-04548-f003:**
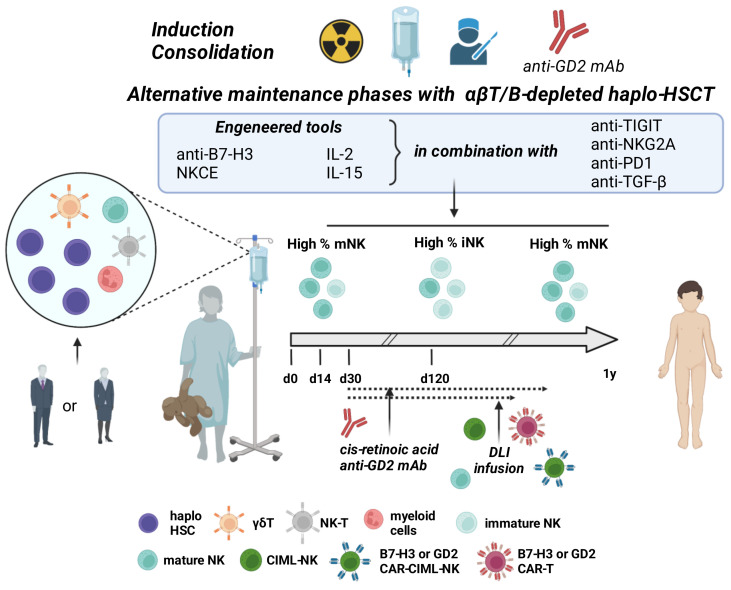
Possible combining strategies to optimize the maintenance phase in HR-NB. AβΤ/B-depleted haplo-HSCT ensure a high number of mature effectors, including NK cells, in the early phase post-transplant. NK cells could be made more effective also using engineered engagers or cytokines modified to reduce their toxicity, eventually in combination with the inhibitory signal blockade. Immune responses against NB could also be potentiated by the infusion of in vitro activated natural or engineered donor T or NK lymphocytes (DLI). CIML-NK = cytokine induced memory-like NK; NKCE = NK cell engagers; CAR = Chimeric antigen receptor. Created with BioRender.com.

**Table 1 cancers-14-04548-t001:** Clinical trials exploring different HSCT and NK cell infusion settings in NB patients. Source: www.clinicaltrials.gov (accessed on 3 August 2022).

Setting	NCT Number	Title	Status	Interventions
	**NCT** **02573896**	Immunotherapy of Relapsed Refractory Neuroblastoma with Expanded NK Cells	Active, not recruiting	Ex-vivo expanded autologous NK cells + standard dosing of anti-GD2 (ch14.18/CHO) mAb + Lenalidomide
**AUTOLOGOUS**	**NCT** **04211675**	NK Cells Infusions With Irinotecan, Temozolomide, and Dinutuximab	Not yet recruiting	Ex-vivo expanded autologous NK Cells + Irinotecan, Temozolomide, and Dinutuximab
	**NCT** **01875601**	NK White Blood Cells and Interleukin in Children and Young Adults with Advanced Solid Tumors	Completed	Cyclophosphamide lymphodepletion + ex-vivo expanded autologous NK cells + rIL15
**HAPLOIDENTICAL**	**NCT** **00698009**	Haploidentical NK Cells in Patients with Relapsed or Refractory Neuroblastoma	Terminated	HLA- Haploidentical NK cell infusion + Interleukin-2
	**NCT** **03242603**	Immunotherapy of Neuroblastoma Patients Using a Combination of Anti-GD2 and NK Cells	Unknown	Ex-vivo expanded activated HLA- Haploidentical NK cells + anti-GD2 (ch14.18/CHO) mAb
	**NCT** **01156350**	Haplo-identical Hematopoietic Stem Cell Transplantation Following Reduced-Intensity Conditioning in Children with Neuroblastoma	Unknown	HLA- Haploidentical HSCT + CD3/CD19 graft depletion
	**NCT** **00877110**	Anti-GD2 3F8 Antibody and Allogeneic Natural Killer cells for High-Risk Neuroblastoma	Completed	HLA-Haploidentical NK cells + anti-GD2 (3F8) mAb
	**NCT** **02650648**	Humanized Anti-GD2 Antibody Hu3F8 and Allogeneic Natural Killer Cells for High-Risk Neuroblastoma	Active, not recruiting	HLA- Haploidentical NK cells + anti-GD2 (humanized 3F8) mAb + rIL-2
	**NCT** **01857934**	Therapy for Children with Advanced Stage Neuroblastoma	Active, not recruiting	anti-GD2 (hu14.18K322A) mAb + HLA- Haploidentical NK cells + rIL-2 + GM-CSF
	**NCT** **02130869**	A Pilot Study of Immunotherapy Including Haploidentical NK Cell Infusion Following CD133+ Positively-Selected Autologous Hematopoietic Stem Cells in Children with High Risk Solid Tumors or Lymphomas	Completed	CD133pos selected autologous stem cell infusion + anti-GD2 (hu14.18K322A) mAb + rIL-2 + HLA-Haploidentical NK cell + G-CSF + GM-CSF
	**NCT** **00790413**	Haploidentical Stem Cell Transplantation in Neuroblastoma	Active, not recruiting	T-cell depleted HLA-Haploidentical HSCT + DLI (T cells) + Rituximab + mesenchymal stem cells
	**NCT** **02258815**	CH14.18 1021 Antibody and IL2 After Haplo SCT in Children with Relapsed Neuroblastoma	Unknown	HLA-Haploidentical HSCT+ anti-GD2 (CH14.18/CHO) mAb + rIL2
	**NCT** **01807468**	Haploidentical Stem Cell Transplantation and NK Cell Therapy in Patients with High-risk Solid Tumors	Unknown	HLA-Haploidentical KIR-L mismatch HSCT + ex-vivo expanded donor-derived NK cells + low-doses of rIL-2
	**NCT** **02100891**	Phase 2 STIR Trial: Haploidentical Transplant and Donor Natural Killer Cells for Solid Tumors	Active, not recruiting	HLA-Haploidentical HSCT + ex-vivo expanded donor-derived NK cells
	**NCT** **01386619**	NK DLI in Patients After Human Leukocyte Antigen (HLA)-Haploidentical Hematopoietic Stem Cell Transplantation (HSCT)	Completed	HLA-Haploidentical HSCT + donor-derived NK cells
	**NCT** **02508038**	Alpha/Beta CD19+ Depleted Haploidentical Transplantation + Zometa for Pediatric Hematologic Malignancies and Solid Tumors	Recruiting	TCR-α/β+ and CD19+ depleted KIR/KIR ligand-mismatched Haploidentical HSCT + zoledronate
	**NCT** **00569283**	Donor Natural Killer Cell Infusion in Preventing Relapse or Graft Failure in Patients Who Have Undergone Donor Bone Marrow Transplant	Completed	HLA-Haploidentical HSCT + donor-derived NK cells
**ALLOGENIC**	**NCT** **01576692**	Combination Chemotherapy, Monoclonal Antibody, and Natural Killer Cells in Treating Young Patients with Recurrent or Refractory Neuroblastoma	Completed	anti-GD2 (Hu14.18K322A) mAb + allogeneic NK cells

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
