# Peer review of "Strategies for Potentiating NK-Mediated Neuroblastoma Surveillance in Autologous or HLA-Haploidentical Hematopoietic Stem Cell Transplants"

_cancers, 2022, doi:10.3390/cancers14194548_

Round 1

Reviewer 1 Report

I congratulate the authors for discussing the current status and proposing new strategies for NK-based therapies for neuroblastoma in a review article titled, " NK Cell-Based Immunotherapies in Refractory/Relapsing 2 High-Risk Neuroblastoma: it is Time for New Strategies ". Overall, the structure of the review is satisfactory. I think it will be great if the authors can include a table summarizing NK- based clinical trials which are currently underway or have been concluded.

Author Response

Best Regards

Reviewer 2 Report

The current review attempts to address issues on NK cell-based therapies for NB. While of interest, the review centers the bulk of summation regarding HSCT application and issues with NK cell correlations and not nearly enough on NK cell themselves. The HSCT procedures and outcomes has numerous components which may or may not reflect on NK cells and their role, much less how to exploit/develop. It is very surprising that no discussion on CAR NK, comparison/contrast with T or CAR T or CAR NK/T, use of bispecifics and now even TriKEs, biology of IL15, even biology of NK cells themselves are not adequately discussed. There is an inordinate amount of discussion on HSCT, KIR/KIR ligand typing, correlations with outcomes but little context of whether this is really where the field is going (even with HSCT for hematologic malignancies this is not having as much of an impact compared to cell engineering areas). There needs to be much less on HSCT and typing discussed and much more on biology of NK (this is also only very superficially described and inaccurate statements such as unlicensed NK cells being anergic are given- there is no evidence for anergy in this population just lesser functionality). In that regard the review is more along HSCT and NB with minimal NK other than KIR/KIR ligand discussed. 

The use adoptively transferred NK, use of IL15 and other cytokines to promote reconstitution, engraftment function (even with other cancers such AML), biology of NK (there is no mention of exhaustion or checkpoint like TIGIT anywhere), NKG2A inhibition, CD16 modulation issues (ie ADAM17 cleavage), cytokines involved in activation (ie IL12, 18, 23), NK cell survival/anergy/exhaustion, NK CAR (and sources), even CAR NK92 targeting CD276 for NB have been reported as well as CAR T approaches targeting other NB antigens such as  CD171 (Abbiasi J JAMA Onc 2021) as well as CAR NK/T with IL15. Even a description on what makes an NK cell versus T versus NK/T important because all being applied.

Author Response

Best Regards

Round 2

Reviewer 2 Report

The review is improved but the title is still misleading in my mind as it really still centers on HSCT approaches and therefore should reflect this. There is still very little real description on what an NK cell and its complex biology which also should be strengthened much more. A table of the receptors/markers expressed and their functions would help with references. Additionally, more on NK biology/activation/development needed especially in context of HSCT where both donor NK cells in the graft and de novo generation and reconstitution kinetics are needed (as well as how diff factors affect). Finally, more depth on why NB is being looked at with NK therapy and how diff than AML and other cancers more often targeted is needed. The cancer biology underlying NB and what makes it unique and different versus other other cancers where the emphasis is on T cells.

Author Response

Dear Reviewer,

Best Regards
